# Immobilization and Release Studies of Triazole Derivatives from Grafted Copolymer Based on Gellan-Carrying Betaine Units

**DOI:** 10.3390/molecules26113330

**Published:** 2021-06-01

**Authors:** Nicolae Baranov, Stefania Racovita, Silvia Vasiliu, Ana Maria Macsim, Catalina Lionte, Valeriu Sunel, Marcel Popa, Jacques Desbrieres, Corina Cheptea

**Affiliations:** 1Department of Natural and Synthetic Polymers, Faculty of Chemical Engineering and Environmental Protection, Gheorghe Asachi Technical University of Iasi, Prof. Dr. Docent Dimitrie Mangeron Street, No. 73, 700050 Iasi, Romania; Baranov_nicolae@yahoo.com; 2Faculty of Chemistry, Al. I. Cuza University, Carol I Bvd., No. 11, 700506 Iasi, Romania; vsunel@uaic.ro; 3Department of “Mihai Dima” Functional Polymers, Petru Poni Institute of Macromolecular Chemistry, Grigore Ghica Voda Alley, No. 41A, 700487 Iasi, Romania; stefania.racovita@icmpp.ro (S.R.); macsim.ana@icmpp.ro (A.M.M.); 4Faculty of Medicine, Gr. T. Popa University of Medicine and Pharmacy, Universitatii Street, No.16, 700115 Iasi, Romania; clionte@yahoo.com; 5Academy of Romanian Scientists, Splaiul Independentei Street No. 54, 050085 Bucuresti, Romania; 6Institut des Sciences Analytiques et de Physico-Chimie pour l’Environnement et les Materiaux (IPREM), Pau and Pays de l’Adour University (UPPA), UMR CNRS 5254, Helioparc Pau Pyrenees, 2 av. President Angot, CEDEX 09, 64053 Pau, France; jacques.desbrieres@univ-pau.fr; 7Department of Biomedical Sciences, Faculty of Biomedical Bioengineering, Gr. T. Popa University of Medicine and Pharmacy, Kogalniceanu Street No. 9–13, 700454 Iasi, Romania; coricheptea@yahoo.com

**Keywords:** grafted copolymer, triazole derivative, adsorption studies

## Abstract

New polymer-bioactive compound systems were obtained by immobilization of triazole derivatives onto grafted copolymers and grafted copolymers carrying betaine units based on gellan and N-vinylimidazole. For preparation of bioactive compound, two new types of heterocyclic thio-derivatives with different substituents were combined in a single molecule to increase the selectivity of the biological action. The 5-aryl-amino-1,3,4 thiadiazole and 5-mercapto-1,2,4-triazole derivatives, each containing 2-mercapto-benzoxazole nucleus, were prepared by an intramolecular cyclization of thiosemicarbazides-1,4 disubstituted in acidic and basic medium. The structures of the new bioactive compounds were confirmed by elemental and spectral analysis (FT-IR and ^1^H-NMR). The antimicrobial activity of 1,3,4 thiadiazoles and 1,2,4 triazoles was tested on gram-positive and gram-negative bacteria. The triazole compound was chosen to be immobilized onto polymeric particles by adsorption. The Langmuir, Freundlich, and Dubinin–Radushkevich adsorption isotherm were used to describe the adsorption equilibrium. Also, the pseudo-first and pseudo-second models were used to elucidate the adsorption mechanism of triazole onto grafted copolymer based on N-vinylimidazole and gellan (PG copolymer) and grafted copolymers carrying betaine units (PGB1 copolymer). In vitro release studies have shown that the release mechanism of triazole from PG and PGB1 copolymers is characteristic of an anomalous transport mechanism.

## 1. Introduction

The design of new bioactive compounds that can be immobilized onto a polymeric support to achieve a controlled/sustained release and to find a way to treat some diseases represents an important goal in medical and pharmaceutical research [1,2].

Depending on the radicals present on the heterocycle, the organic compounds that contain in their molecule 1,3,4-thiadiazole and 1,2,4-triazole heterocycles have gained a remarkable interest due to their different biological properties, such as: antibacterial [3,4,5], antifungal [6,7,8,9], tuberculostatic [10,11], analgesic [12,13], anti-inflammatory [14,15], hypnotic and sedative [16], cardioprotective [17], anticancer [18,19,20,21,22], anti-ulcerative [23], anticonvulsant [24], antidiabetic agents [25], allosteric modulators [26], and cathepsin B and tubulin inhibitors [27,28].

At the same time, the addition of a benzoxazole structure can be useful for the improvement of the biological activity. An argument in favor of this statement is that the compounds with benzoxazole heterocycle in their structure have practically the same biological properties like thiadiazole and triazole derivatives, such as antimicrobial, antifungal, antiviral, anti-inflammatory, antipyretic, analgesic, anticonvulsant, tuberculostatic, and antitumor agents [29,30,31,32,33,34,35,36]. Benzoxazole and triazole heterocyclic rings played an important role in the improving of antimicrobial and anticancer activity of the bioactive compounds [37].

These facts served as an argument in the synthesis of new heterocyclic compounds containing 1,3,4-thiadiazole and 1,2,4-triazole nucleus and the rest of 2-mercaptobenzoxazole, considering their mutual influence as well as the biological effect that can be generated throughout the molecule.

Although the researches on compounds with a benzoxazole structure are multiple and varied [38,39,40,41], some aspects of the synthesis and studies of these types of compounds in which both thiadiazole and triazole structures are grafted on the aromatic heterocycle, 2-mercaptobenzoxazole, have not been studied. Based on this premise, we expanded our research to the synthesis of new derivatives of 1,3,4-thiadiazole and 1,2,4-triazole classes of compounds using 1,4-disubstituted thiosemicarbazides as intermediates.

Also, the choice of macromolecular support for immobilization of bioactive compounds is very important. Encouraged by the preliminary results obtained in our previous study [42], we chose to use for the immobilization of these new bioactive compounds two types of macromolecular supports: grafted copolymers and grafted copolymers carrying betaine units based on gellan and N-vinylimidazole (PG and PGB1, respectively). The structure of PG and PGB1 copolymers are presented in Figure 1.

In this paper, our investigations were directed to the following directions:Synthesis of new bioactive compounds in which thiadiazole and triazole structures are grafted onto the aromatic heterocycle, 2-mercaptobenzoxazole, followed by the choice of the compound with the best biological properties in order to be immobilized on a polymeric support;Synthesis of new polymer-bioactive compound systems; andDetailed studies on the immobilization of the bioactive compound by adsorption on grafted copolymers and those containing the betaine structure.

## 2. Results and Discussion

### 2.1. Synthesis of Thiadiazole and Triazole Derivatives

Among the heterocyclic combinations, the 1,3,4-thiadiazole and 1,2,4-triazole derivatives present a remarkable application interest. The synthesis of 1,3,4-thiadiazole and 1,2,4-triazole derivatives were performed in several steps as follows:

Step 1. Synthesis of 1-(benzoxazole-2′-yl-mercapto-acetyl)-4-aryl-thiosemicarbazides (II–VII) was performed in methanol solution by heating a mixture of benzoxazolyl-2-mercapto-acetic acid hydrazide (I) [43] with various types of isothiocyanate derivatives: phenyl, p-tolyl, p-methoxyphenyl, p-bromophenyl, p-chlorophenyl, and p-iodofenil, according to the procedures described in our previous papers for other types of thiosemicarbazides (Figure 2) [4,44,45].

The structure elucidation of the compounds II–VII was carried out by elemental analysis as well as by spectroscopic methods (FT-IR and ^1^H-NMR).

The results obtained from the elemental analysis as well as the chemical formula and the yield of reaction of all bioactive compounds synthesized in this paper (compounds II-XIX) are presented in Table 1.

In the FT-IR spectra (Figure not shown), the following characteristic absorption bands can be observed: 700–749 cm^−1^ assigned to the C-S stretching vibration; 1400–1498 cm^−1^ attributed to the CH_2_ bending vibration; 1620–1704 cm^−1^ assigned to the C=O stretching vibrations; the C=S stretching vibration was observed at 1120–1150 cm^−1^; NH stretching vibration appears at 2880–3150 cm^−1^; the absorption bands of the mono and disubstituted aromatic rings were identified in the region of 680–789 cm^−1^; and the C-Br, C-Cl, and C-I stretching vibrations appear at 629 cm^−1^, 725 cm^−1^, and 630 cm^−1^, respectively.

In the ^1^H-NMR spectra, the two methylene protons generate a singlet at δ = 4.25–4.60 ppm and at δ = 8.67–8.89 ppm, while the protons from the -NH group appear at 10.06–10.85 ppm. The aromatic protons in phenyl and benzoxazole appear within the range 6.50–7.92 ppm. The protons belonging to the methyl groups are identified at δ = 2.11 ppm for the compound III and at δ = 3.38 ppm for the thiosemicarbazide IV (Figure 3).

Step 2. Synthesis of new 1,3,4-thiadiazoles and 1,2,4-triazoles starting from 1,4-disubstituted-thiosemicarbazides (compounds II–VII). The 5-aryl-amino-2-substituted-1,3,4-thiadiazoles (compounds VIII–XIII) were obtained by intramolecular cyclization in concentrated sulfuric acid medium, while the 5-mercapto-3,4-disubstituted -1,2,4-triazoles (compounds XIV–XIX) were prepared under the catalytic action of hydroxyl ions, according to our previously work described in literature (Figure 4) [4,44,45,46,47].

The idea of synthesis of grafted 1,3,4-thiadiazoles on the 2-mercapto-benzoxazole molecule is an extremely interesting possibility to find new derivatives, which can be active against some microbial strains.

The mechanism of the synthesis reaction of these types of heterocyclic compounds is presented in Figure 5 and contains the following steps:

In case of 1,2,4-triazoles, the mechanism of cyclization in the presence of hydroxyl ions (Figure 6) begins with the hydrogen extraction from the nitrogen atom situated at position 4 of 1,4-disubstituted thiosemicarbazide, which has an increased mobility due to the p-π conjugation between the lone pair electrons belonging to the nitrogen atom and the π electrons of thionic bond or of benzene nucleus, respectively. As a result of this conjugation, a negative charge appears on the nitrogen atom from position 4 that can be in equilibrium with another anion formed at the sulfur atom. In this mesomer ion, the negatively charged nitrogen atom (more stable) attacks the carbon belonging to the carbonyl group simultaneously with π electron delocalization, leading to the formation of the five-membered heterocycle, the negative charge being located at the oxygen atom outside the ring. This structure is stabilized by eliminating the hydroxyl group in the reaction medium leading to the formation of 5-mercapto-1,2,4-triazole structure. Depending on the preparation conditions and the reactant nature, as well as due to the presence of thioamide group in triazole molecule, 5-mercapto-3,4 disubstituted-1,2,4-triazoles show the phenomenon of double reactivity, reacting as if they had a thionic structure (A) or a thiol structure (B). This behavior explains the presence of a reaction center at the sulfur atom from 5-position of five-membered heterocycle [48].

In order to obtain the information about the structure of 1,3,4-thiadiazoles (compounds VIII–XIII) and 1,2,4-triazoles (compounds XIV–XIX), the elemental analysis (Table 1) as well as the FT-IR (Figure 7) and ^1^H-NMR spectra (Figure 8) were performed.

Generally, in FT-IR spectra of the 1,3,4-thiadiazole (compounds VIII–XIII), the following absorption bands can be observed: the absorption bands at 3318–2880 cm^−1^ corresponding to the NH stretching vibrations; the absorption band at 1438–1462 cm^−1^ can be attributed to the stretching vibration of -N=C-S group; the bending vibration of the CH_2_ group has been assigned to the absorption bands situated between 1401–1496 cm^−1^; characteristic CH bands from mono and disubstituted benzene nucleus are assigned to the absorption band situated in the range of 680–789 cm^−1^; and the C-Br, C-Cl, and C-I stretching vibrations appear at 629, 725, and 630 cm^−1^, respectively.

In the case of 1,2,4-triazole derivatives, the FT-IR spectra of the compounds XIV–XIX present a wide band at 3553 cm^−1^ due to the NH stretching vibrations; 1613–1628 cm^−1^ corresponding to the C=N stretching vibration; the C=S group appears at 1490–1495 cm^−1^ while the bands at 837–942 cm^−1^ can be attributed to the para-disubstituted benzene nucleus; the S-CH_2_ stretching vibration has been assigned to an adsorption band at 765–785 cm^−1^; and the C-Br, C-Cl, and C-I stretching vibrations show maximum adsorption bands at 753, 746, and 748 cm^−1^, respectively.

For example, in Figure 7, the spectra of compound VIII (a) and compound XVI (b), respectively, are presented.

^1^H-NMR spectra bring additional arguments regarding the structure of the obtained organic compounds. In case of 1,3,4-thiadiazole derivatives, the protons of the CH_2_ group bound to the sulfur atom situated at position 2′ of the benzoxazole nucleus appear as a singlet in the region of 4.25–4.53 ppm. The proton bound to the nitrogen could be detected as a singlet in the region 10.27–10.98 ppm. For all types of 1,3,4-thiadiazoles, aromatic protons appear at 6.91–7.56 ppm and 8.21–8.57 ppm, respectively. The protons of the methyl group in 1,3,4-thiadiazoles (compounds IX and X) are found as a triplet at 2.98 and 3.38 ppm, respectively.

The ^1^H-NMR spectra of 1,3,4-thiadiazoles (compound VIII) and 1,2,4-triazoles (compound XVI) are presented in Figure 8.

The ^1^H-NMR spectra of the 1,2,4-triazole derivatives (compounds XIV–XIX) present a singlet due to the protons of the CH_2_ group at 4.17–4.18 ppm, while at 14.04–14.33 ppm is observed a singlet specific to the proton belonging to the NH group. For all 1,2,4-triazoles, the aromatic protons appear at 6.99–7.85 ppm, and for compounds XV and XVI, the protons of the methyl group appear at 2.43 and 3.43 ppm, respectively.

### 2.2. Antibacterial Activities of Bioactive Compounds

The antimicrobial activity of 1,3,4 thiadiazoles and 1,2,4 triazoles was tested according to the National Committee for Clinical Laboratory Standards Institute NCCLS Approval Standard Document M2-A5, Vilanova, PA, USA (2000), on the following microbial strains: gram-positive bacteria (*Staphylococcus aureus* (ATCC-2593), *Bacillus subtilis* (ATCC-6638), *Bacillus cereus* (ATCC-10876)) and gram-negative bacteria (*Salmonella enteritidis* (P-1131) and *Escherichia coli* (ATCC-25922)).

The results regarding the antimicrobial activity of the bioactive compounds synthesized can be seen in Table 2.

Experimental data shows that 1,3,4-thiadiazoles (compounds VIII–XIII) are sensitive to *Staphyloccocus aureus* and present a moderate effect on *Bacillus subtilis* and *Escherichia coli*, while the culture of *Bacillus cereus* and *Salmonella enteritidis* are resistant to the action of the 1,3,4-thiadiazole test.

The 1,2,4-triazoles (compounds XIV–XVI) have been shown to be strong inhibitors on the growth of the bacteria *Escherichia coli*, *Salmonella enteritidis*, and *Bacillus subtilis* in any concentration and regardless of the contact time, while against *Staphylococcus aureus* and *Bacillus cereus* germs, the 1,2,4-triazoles show a moderate action. In contrast, the 1,2,4-triazoles (compounds XVII–XIX) show a moderate action on all tested germs only in a short period of contact time and at all concentrations. It can be stated that the tested 1,3,4-thiadiazoles and 1,2,4-triazoles show appreciable antimicrobial activity against microbial strains similar to that of the model drug, kanamycin.

### 2.3. Immobilization Studies of Triazoles

Among the new synthesized compounds, the 1,2,4-triazole derivative obtained with the highest yield (88%) and the best antimicrobial activity, namely 3-(benzoxazole-2′-yl-mercapto-methyl)-4-(p-methoxyphenyl)-5-mercapto-1,2,4-triazole (compound XVI), which was selected for immobilization by sorption onto grafted copolymers.

The interaction between PG and PGB1 copolymers and compound XVI was highlighted by FT-IR spectroscopy (Figure 9).

For PG—1,2,4-triazole systems (PG-T), it can be observed some new absorption bands as follows: the absorption band at 3140 cm^−1^ due to the stretching vibration of -NH group; 1542 cm^−1^ assigned to the stretching vibration of C=S group; 1450 cm^−1^ corresponding to the stretching vibration of -N=C-S group; and 793 cm^−1^ assigned to the stretching vibration of S-CH_2_ group.For PGB1—1,2,4-triazole systems (PGB1-T), the changes that occur in the FTIR spectrum are the following: the band vibration of 3439 cm^−1^ from PGB1 copolymer is shifted in PGB1-T spectra to 3445 cm^−1^ due to the overlap of the absorption band of -C=N^+^ group from PGB1 copolymer, with absorption band corresponding to the -NH group belonging to the 1,2,4-triazole; absorption band at 832 cm^−1^ is assigned to the stretching vibration of C-H out of plane bending.

Also, XRD and SEM studies were performed for a better characterization of PG-T and PGB1-T systems. The XRD patterns of 1,2,4-triazole, PG-T, and PGB1-T are presented in Figure 10.

The XRD analysis of PG-T and PGB1-T systems showed the combined signals of both PG/PGB1 copolymers and 1,2,4-triazole, leading to the conclusion that the 1,2,4-triazole was successfully immobilized onto polymeric supports.

Visualization of surface morphology of PG-T and PGB1-T systems was achieved through SEM microscopy with a magnification of 1000, and the images are presented in Figure 11.

From SEM images, it can be observed that the PG-T and PGB1-T systems appear to have porous structures different from those of PG and PGB1 copolymers. The changes in the surface morphologies of PG-T and PGB1-T systems indicate that the sorption of 1,2,4-triazole onto polymeric support occurred.

Generally, in order to evaluate a sorption process of bioactive compound, it is important to consider two physico-chemical aspects, namely the equilibrium and the kinetics of sorption.

#### 2.3.1. Sorption Isotherms

To describe the interactions between sorbate and sorbent that occurs in sorption processes, three model isotherms were used, namely: Langmuir, Freundlich, and Dubinin–Radushkevich.

Langmuir isotherm

The Langmuir isotherm describes the sorption in a homogeneous system [49] and is given by the equation:(1)$$q_{e}=\frac{q_{m}\cdot K_{L}\cdot C_{e}}{1+K_{L}\cdot C_{e}}$$
where: ***q_e_*** represents the amount of 1,2,4-triazole sorbed at equilibrium (mg/g); ***q_m_*** represents the maximum amount sorbed (mg/g); ***C_e_*** is the concentration of 1,2,4-triazole solution at equilibrium (mg/L); and ***K_L_*** is the Langmuir constant that reflects the affinity between sorbate and sorbent.

To determine whether the sorption system used is favorable or unfavorable to the sorption, the equilibrium parameter ***R_L_***, given by the equation [50], was also calculated:(2)$$R_{L}=\frac{1}{1+K_{L\cdot }C_{i}\;}\;$$
where ***C_i_*** represents the initial concentration. According to the literature [51], if ***R_L_*** < 1, then the sorption is unfavorable, for ***R_L_*** = 1, the sorption is linear; if 0 < ***R_L_*** < 1, the sorption is favorable or irreversible if ***R_L_*** = 0.

Freundlich isotherm

Another model used in sorption studies is the Freudlich isotherm. This isotherm model is applied in the case of multilayer sorption of sorbate on a heterogeneous surface [52]. The Freundlich isotherm is described by the following equation:(3)$$q_{e}=K_{F}\cdot C_{e}^{1/n_{f}}\;$$
where ***K_F_*** is the Freundlich constant which represents the amount of 1,2,4-triazole sorbed per gram of sorbent when the equilibrium concentration is equal to unity (L/g); 1/***n_f_*** indicates the type of isotherm as follows: favorable 0 < 1/***n_f_*** < 1 or unfavorable 1/***n_f_*** >1.

Dubinin–Radushkevich isotherm

The Dubinin–Radushkevich isotherm is an empirical model designed to estimate the apparent free energy of sorption as well as to differentiate between the physical and chemical sorption process [53]. The Dubinin–Radushkevich model equation is given by the following mathematical relation:(4)$$q_{e}=q_{DR}\cdot exp\left\{-\beta {\left[RTln\left(1+\frac{1}{C_{e}}\right)\right]}^{2}\right\}\;$$
where ***q_DR_*** represents the maximum amount sorbed (mg/g); ***β*** is the Dubinin–Radushkevich constant (mol^2^/kJ^2^); ***R*** represents the ideal gas constant (***R*** = 8.314 kJ/mol⋅K); and ***T*** is the temperature (K).

The Dubinin–Radushkevich isotherm constant, ***β***, is associated with the average free sorption energy, ***E*** (kJ/mol), calculated using the following equation:(5)$$E=1⁄{\left(2\beta \right)}^{\frac{1}{2}}\;$$

The value of ***E*** is used to obtain information about the nature of the sorption process. The sorption process can be physical when the ***E*** values are between 1 and 8 kJ/mol, ion exchange for values of ***E*** between 8 and 16 kJ/mol, and chemical nature for values of ***E*** greater that 16 kJ/mol [54].

The parameters of the studied isotherms as well as the values of the error functions ***R***^2^ and ***χ***^2^ are presented in Table 3.

From data presented in Table 3, the following conclusions can be drawn:The theoretical values obtained for the maximum sorption capacity (***q_m_***) calculated on the basis of the Langmuir isotherm are close to the experimental values ***q_c_*** (457, 512 and 618 mg 1,2,4-triazole /g PG copolymer and 529, 601 and 672 mg of 1,2,4-triazole /g PGB1 copolymer);The values of the equilibrium parameter, ***R_L_***, were in the range between 0 and 1, thus confirming that the PG and PGB1 copolymers are favorable supports for the sorption of 1,2,4-triazoleat at the three temperatures studied. It is also observed that the ***K_L_*** values are higher in the case of the PGB1 copolymer than in the case of the PG copolymer, which indicates a higher affinity of the PGB1 copolymer for 1,2,4-triazole; this is in agreement with the highest sorption capacity obtained in the case of the PGB1 copolymer;The values for ***R***^2^ and ***χ***^2^ are in the range of 0992–0.997 and 2.010–3.429, respectively, indicating that the Langmuir isotherm describes well the experimental data;Although the values of 1/***n_f_*** are in the range of 0–1, which indicates that the Freundlich isotherm is favorable in the case of 1,2,4-triazolesorption onto PG and PGB1 copolymers, the small values of ***R***^2^ (0.911 to 0.921) associated with high values of ***χ***^2^ (18.782–38.332) shows that the Freundlich isotherm does not describe well the experimental data;The analysis of the parameter values obtained by applying the Dubinin–Radushkevich isotherm shows that the ***q_DR_*** values are very close to the experimental values, which indicates that this isotherm describes well the experimental data. This is also supported by the fact that high values for ***R***^2^ (0.997–0.999) and low values for ***χ***^2^ (0.226–0.413) were obtained, confirming that the Dubinin–Radushkevich isotherm describes very well the sorption of 1,2,4-triazole onto PG and PGB1 copolymers;The calculated values of the average free energy of 1,2,4-triazole sorption onto PG and PGB1 copolymers are in the range of 1.099–3.714 kJ/mol, which indicates that the sorption process studied is physical in nature.

#### 2.3.2. Thermodynamic Parameters

By means of thermodynamic parameters, such as Gibbs free energy change (Δ***G***), enthalpy change (Δ***H***), and entropy change (Δ***S***), the mechanism and the type of sorption process can be determined. Thus, according to the literature [55], it is known that in the case of physical sorption, the values of Δ***H*** are less than 40 kJ/mol, while in the case of chemical sorption, the values of Δ***H*** are in the range of 40–120 kJ/mol. The value of Δ***H*** and Δ***S*** can be determined by means of the Langmuir constant using the Van ’t Hoff equation [56]:(6)$$lnK_{L}=\frac{\Delta S}{R}-\frac{\Delta H}{RT}\;$$

From the linear representation ln ***K_L_*** versus 1/***T*** (Figure 12), the values of Δ***S*** and Δ***H*** were obtained from the intercept and slope, respectively, and the results are presented in Table 4. The values of Δ***G*** were obtained using the following equation:(7)ΔG=ΔH−TΔS 

As it can be seen from Table 4, the negative values of Δ***G*** show that the 1,2,4-triazole sorption onto PG and PGB1 copolymers is spontaneous and favorable. Also, the values of this thermodynamic parameter decrease with increasing of the temperature, indicating that there is a higher efficiency of the sorption process at higher temperatures. The values of Δ***H*** < 40 kJ/mol indicate that the interactions between the PG or PGB1 copolymers and 1,2,4-triazole are physical in nature. On the other hand, the positive value of the Δ***H*** shows that the sorption process is endothermic and the increase of the temperature leads to an increase in the amount of 1,2,4-triazole sorbed. The positive value of Δ***S*** indicates the affinity of PG and PGB1 copolymers for 1,2,4-triazole as well as the increase randomness at the solid-solution interface during the sorption process [57]. Also, the positive value of Δ***S*** can indicate an increase in the degree of freedom of 1,2,4-triazole. Moreover, the positive values of Δ***H*** and Δ***S*** lead to the conclusion that the sorption process occurs spontaneously at all temperatures.

#### 2.3.3. Sorption Kinetic Study

The study of sorption kinetic describes the rate of sorption and is very important because it gives us information about the mechanism of sorption. Two mathematical models, such as the Lagergren model (pseudo-first order kinetic model) and the Ho model (pseudo-second order kinetic model), were used to elucidate the mechanism of 1,2,4-triazole sorption onto PG and PGB1 copolymers. The two models mentioned above are described by the following equations:Lagergren model [58]:
(8)$$q_{t}=q_{e}\left(1-e^{-k_{1}t}\right)\;$$

Ho model [59]:

(9)$$q_{t}=\frac{k_{2}\cdot q_{t}^{2}\cdot t}{1+k_{2}\cdot q_{e}\cdot t}\;$$
where ***q_e_*** and ***q_t_*** are the amounts of 1,2,4-triazole sorbed at equilibrium and at time ***t*** (mg/g), ***k***_1_ is the rate constant of the pseudo-first order sorption process (min^−1^), and ***k***_2_ is the rate constant of pseudo-second order sorption process (g/mg⋅min).

Figure 13 presents the plots of the Lagergren and Ho models in the case of 1,2,4-triazole sorption (C_1,2,4-triazole_ = 15 × 10^−3^ g/mL) onto PG and PGB1 copolymers at ***T*** = 308 K, and Table 5 shows the parameter values corresponding to the Lagergren and Ho models.

From data presented in Table 5, it is observed that the value of the sorption rate increases with the increase of the 1,2,4-triazole concentration. The highest amount of 1,2,4-triazole sorbed was obtained in the case of the PGB1 copolymer. The ***q_e_*** values calculated by applying the Lagergren model are close to the experimental values compared to the ***q_e_*** values calculated by applying the Ho model. Although the values of the correlation coefficients ***R***^2^ are higher when applying the two kinetic models, the ***χ***^2^ values are higher when applying the Ho model, which indicates that the Lagergren model describes better the experimental data. These results indicate that 1,2,4-triazole sorption onto PG and PGB1 copolymer is a physical process.

### 2.4. In Vitro Release Studies

In vitro release studies were performed on the PG-T and PGB1-T systems with the highest amount of immobilized 1,2,4-triazole. In vitro 1,2,4-triazole release studies were realized at pH = 1.2, and the release profiles are shown in Figure 14.

The kinetic studies and the release mechanism from PG-T and PGB1-T copolymers loaded with 1,2,4-triazole were examined using the following mathematical models:Higuchi model [60]:
(10)$$Q_{t}=k_{H}\cdot t^{1/2}\;$$
where ***Q_t_*** is the amount of drug released at time ***t***; ***k_H_*** is the Higuchi dissolution constant; and ***t*** is the time.

Korsmeyer–Peppas model [61]:

(11)$$\frac{M_{t}}{M_{\infty }}=k_{r}\cdot t^{n}\;$$
where ***M_t_***/***M***_∞_ is the fraction of drug released at time ***t***; ***k_r_*** is the release rate constant that is characteristic to bioactive compound-polymer interactions; and ***n*** is the diffusion coefficient that is characteristic to the release mechanism. The values of the 1,2,4-triazole release parameters from PG-T and PGB1-T systems are presented in Table 6.

The release exponent n from the Korsmeyer–Peppas equation is situated between 0.534 and 0.634. These results suggest that the release mechanism of 1,2,4-triazole from PG-T and PGB1-T systems was controlled by more than one process, namely diffusion and swelling.

## 3. Materials and Methods

### 3.1. General Information

All reactants and solvents were purchased from Merck Company KGaA (Darmstadt, Germany) and were used without purification. Elemental analyses were performed by Exeter Analytical CF-440 Elemental Analyzer (Coventry, United Kingdom). The bioactive compounds (1,4-disubstituted-thiosemicarbazides, 1,3,4-thiadiazoles, and 1,2,4-triazoles) were characterized by FT-IR spectroscopy (Bruker Vertex FT-IR spectrometer) as potassium bromide pellets in the range of 4000–400 cm^−1^ and at a resolution of 2 cm^−1^. The ^1^H NMR spectra were recorded in DMSO-d_60_ on Bruker ARX 400 Spectrometer (400 MHz). X-ray diffraction analysis was performed in a Rigaku Miniflex 600 diffractometer using CuKα-emission in the angular range 3–90° (2θ) with a scanning step of 0.01° and a recording rate of 5°/min. The surface morphologies of PG-T system, PGB1-T system, and 1,2,4-triazole in powder form were analyzed with an environmental scanning electron microscope type Quanta 200 at 25kV with secondary electrons in low vacuum.

### 3.2. Synthesis of Benzoxazole-2′-yl-mercapto-acetic Acid Hydrazide (Compound I)

It was obtained by treating benzoxazol-2′-yl-mercapto-acetic acid ethyl ester with 98% hydrazine hydrate in anhydrous ethanol at room temperature.

### 3.3. General Procedure for Synthesis of 1-(Benzoxazole-2′-yl-mercapto-acetyl)-4-aryl-thiosemicarbazide (Compounds II–VII)

A total of 0.005 moles of benzoxazole-2′-yl-mercapto-acetic acid hydrazide (I) dissolved in 30 mL absolute methyl alcohol was treated with 0.005 moles of aromatic isothiocyanate in 5 mL anhydrous methanol. The mixture was refluxed for 2 h. During heating, a crystalline precipitate appeared and became more and more abundant once the reaction was complete. After cooling, the precipitate was filtered and washed with anhydrous ethyl ether. Purification was carried out by recrystallization from anhydrous methanol.

*1-(Benzoxazole-2′-yl-mercapto-acetyl)-4-phenyl-thiosemicarbazide* (Compound II), Physical state: White, crystalline powder; M_p_ = 182–183 °C. FT-IR (KBr,ν, cm^−1^): 2917, 3150 (NH); 1670 (C=O); 1400 (CH_2_); 1145 (C=S); 755 (monosubstituted benzene nucleus); 700 (C-S).

*1-(Benzoxazole-2′-yl-mercapto-acetyl)-4-(p-tolyl)-tiosemicarbazide* (Compound III), Physical state: White-ivory powder, acicular crystals; M_p_ = 163–164 °C. FT-IR (KBr,ν, cm^−1^): 2880, 3096 (NH); 1709 (C=O); 1424 (CH_2_); 720 (C-S); 750 (para-disubstituted benzene nucleus). ^1^H-NMR (DMSO-*d*_6_, 400 MHz, δ ppm): 2.11 (t, 3H, CH_3_); 4.39 (s, 2H, CH_2_); 6.95–6.98 (d, 2H, CHAr); 7.13–7.18 (d, 2H, CHAr); 7.57–7.63 (d, 2H, CHAr); 7.76–7.79 (d, 2H, CHAr); 9.86 (s, 1H, NHCO); 10.19 (s, 1H, NH); 10.76 (s, 1H, NH).

*1-(Benzoxazole-2′-yl-mercapto-acetyl)-4-(p-methoxytolyl) thiosemicarbazide* (Compound IV), Physical state: White, crystalline powder; M_p_ = 186–188 °C. FT-IR (KBr,ν, cm^−1^): 2918, 3096 (NH); 1623 (C=O); 1492 (CH_2_); 1239 (C=S); 789 (monosubstituted benzene nucleus); 749 (C-S). ^1^H-NMR (DMSO-*d*_6_, 400 MHz, δ ppm): 3.38 (t, 3H, OCH_3_); 4.38 (s, 2H, CH_2_); 6.09–6.11 (d, 2H, CHAr); 7.11–7.13 (d, 2H, CHAr); 7.41–7.43 (d, 2H, CHAr); 7.84–7.92 (d, 2H, CHAr); 8.86 (s, 1H, NHCO); 10.26 (s, 1H, NH); 10.86 (s, 1H, NH).

*1-(Benzoxazole-2′-yl-mercapto-acetyl)-4-(p-chlorophenyl) thiosemicarbazide* (Compound V), Physical state: White-ivory, crystalline powder; M_p_ = 147–149 °C. FT-IR (KBr,ν, cm^−1^): 3096 (NH); 1620 (C=O); 1482 (CH_2_); 1249 (C=S); 749 (para-disubstituted benzene nucleus); 710 (C-S); 629 (C-Br). ^1^H-NMR (DMSO-*d*_6_, 400 MHz, δ ppm): 4.27 (d, 2H, CH_2_); 7.32–7.35 (d, 2H, CHAr); 7.38–7.40 (d, 2H, CHAr); 7.52–7.56 (d, 2H, CHAr); 7.65–7.67 (d, 2H, CHAr); 9.67 (s, 1H, NHCO); 9.90 (s, 1H, NH); 10.49 (s, 1H, NH).

*1-(Benzoxazole-2′-yl-mercapto-acetyl)-4-(p-chlorophenyl)-thiosemicarbazide* (Compound VI), Physical state: White-ivory, crystalline powder; M_p_ = 150–151 °C. FT-IR (KBr,ν, cm^−1^): 3080, 3124 (NH); 1620 (C=O); 1498 (CH_2_); 1250 (C=S); 725 (C-Cl); 708 (C-S); 680 (para-disubstituted benzene nucleus). ^1^H-NMR (DMSO-*d*_6_, 400 MHz, δ ppm): 4.23 (d, 2H, CH_2_); 6.92–7.04 (d, 2H, CHAr); 7.29–7.32 (d, 2H, CHAr); 7.39–7.41 (d, 2H, CHAr); 8.77 (s, 1H, NHCO); 10.11 (s, 1H, NH); 10.85 (s, 1H, NH).

*1-(Benzoxazole-2′-yl-mercapto-acetyl)-4-(p-iodophenyl)-thiosemicarbazide* (Compound VII). Physical state: White-yellow, crystalline powder; M_p_ = 172–179 °C. FT-IR (KBr,ν, cm^−1^): 3097, 3140 (NH); 1620 (C=O); 1490 (CH_2_); 1120 (C=S); 785 (para-disubstituted benzene nucleus); 712 (C-S); 630 (C-I). ^1^H-NMR (DMSO-*d*_6_, 400 MHz, δ ppm): 4.25 (d, 2H, CH_2_); 6.91–7.03 (d, 2H, CHAr); 7.30–7.31 (d, 2H, CHAr); 7.40–7.42 (d, 2H, CHAr); 7.66–7.69 (d, 2H, CHAr); 8.74 (s, 1H, NHCO); 10.10 (s, 1H, NH); 10.31 (s, 1H, NH).

### 3.4. General Procedure for Synthesis of 1-(Benzoxazole-2′-yl-mercapto-methyl)-5-(aryl-amino)-1,3,4,thiadiazoles (Compounds VIII–XIII)

A total of 6 mL of concentrated sulfuric acid were added under stirring to 0.005 moles of 1-(benzoxazole-2′-yl-mercapto-acetyl)-4-aryl-thiosemicarbazide (compounds II–VII). The reaction mixture was stirred at room temperature for 45–50 min to complete the cyclization and then was poured on crushed ice when an abundant precipitate appeared. The mixture was neutralized with ammonium hydroxide, and, after 2 h, the microcrystals were filtered in vacuum and washed with distilled water until the washing water reached pH = 7. Then, the obtained compounds were dried in vacuum oven at 55–60 °C and were purified by recrystallization from ethyl alcohol.

*2-(Benzoxazole-2′-yl-mercapto-methyl)-5-(phenyl-amino)-1,3,4-thiadiazole* (Compound VIII), Physical state: White-yellow, crystalline powder; M_p_ = 194–196 °C. FT-IR (KBr,ν, cm^−1^): 2945, 3256 (NH); 1496 (CH_2_); 1454 (N=C-S); 1401, 1535 (thiadiazole nucleus) 1090 (C-S-C); 785, 815 (para-disubstituted benzene nucleus); 750 (C-S). ^1^H-NMR (DMSO-*d*_6_, 400 MHz, δ ppm): 4.26 (s, 2H, CH_2_); 6.91 (t, 1H, CHAr); 6.98–7.00 (d, 2H, CHAr); 7.46–7.49 (d, 2H, CHAr); 8.35 (d, 2H, CHAr); 8.71–8.72 (d, 2H, CHAr); 10.48 (s, 1H, NH).

*2-(Benzoxazole-2′-yl-mercapto-methyl)-5-(tolyl-amino)-1,3,4-thiadiazole* (Compound IX), Physical state: Brown-red, amorphous powder; M_p_ = 154–156 °C. FT-IR (KBr,ν, cm^−1^): 2945 (NH); 1480 (CH_2_); 1450 (N=C-S); 1448, 1517 (thiadiazole nucleus); 1070 (C-S-C); 828 (para-disubstituted benzene nucleus); 730 (C-S). ^1^H-NMR (DMSO-*d*_6_, 400 MHz, δ ppm): 2.29 (t, 3H, CH_3_); 4.49 (s, 2H, CH_2_); 7.36–7.38 (d, 2H, CHAr); 7.66–7.69 (d, 2H, CHAr); 8.21–8.23 (d, 2H, CHAr); 8.39–8.42 (d, 2H, CHAr); 10.21 (s, 1H, NH).

*2-(Benzoxazole-2′-yl-mercapto-methyl)-5-(p-methoxyphenyl-amino)-1,3,4-thiadiazole* (Compound X), Physical state: Brown, crystalline powder; M_p_ = 116–118 °C. FT-IR (KBr,ν, cm^−1^): 2500, 3000 (NH); 1475 (CH_2_); 1438 (N=C-S); 1243, 1615 (thiadiazole nucleus); 1068 (C-S-C); 830 (para-disubstituted benzene nucleus); 710 (C-S). ^1^H-NMR (DMSO-*d*_6_, 400 MHz), δ ppm: 3.38 (t, 3H, OCH_3_); 4.53 (s, 2H, CH_2_); 7.31–7.34 (d, 2H, CHAr); 7.53–7.56 (d, 2H, CHAr); 8.50 (d, 2H, CHAr); 8.86 (d, 2H, CHAr); 10.37 (s, 1H, NH).

*2-(Benzoxazole-2′-yl-mercapto-methyl)-5-(p-bromophenyl-amino)-1,3,4-thiadiazole* (Compound XI), Physical state: Brown, crystalline powder; M_p_ = 197–197°C. FT-IR (KBr,ν, cm^−1^): 2880, 2905 (NH); 1480 (CH_2_); 1449 (N=C-S); 1240, 1518 (thiadiazole nucleus); 1070 (C-S-C); 828 (para-disubstituted benzene nucleus); 750 (C-Br). ^1^H-NMR (DMSO-*d*_6_, 400 MHz), δ ppm: 4.49 (s, 2H, CH_2_); 7.21-7.23 (d, 2H, CHAr); 7.47-7.51 (d, 2H, CHAr); 8.19-8.21 (d, 2H, CHAr); 8.33-8.35 (d, 2H, CHAr); 10.98 (s, 1H, NH).

*2-(Benzoxazole-2′-yl-mercapto-methyl)-5-(p-chlorophenyl-amino)-1,3,4-thiadiazole* (Compound XII), Physical state: Brown-white, crystalline powder; M_p_ = 201–203 °C. FT-IR (KBr,ν, cm^−1^): 3318 (NH); 1475 (CH_2_); 1450 (N=C-S); 1240, 1518 (thiadiazole nucleus); 1071 (C-S-C); 830 (para-disubstituted benzene nucleus); 755 (C-Cl). 1H-NMR (DMSO-*d*_6_, 400 MHz), δ ppm: 4.32 (s, 2H, CH_2_); 7.17–7.19 (d, 2H, CHAr); 7.39–7.42 (d, 2H, CHAr); 8.22–8.24 (d, 2H, CHAr); 8.54–8.57 (d, 2H, CHAr); 10.31 (s, 1H, NH).

*2-(Benzoxazole-2′-yl-mercapto-methyl)-5-(p-iodophenyl-amino)-1,3,4-thiadiazole* (Compound XIII), Physical state: Brown, crystalline powder; M_p_ = 187 °C. FT-IR (KBr,ν, cm^−1^): 2880, 2918 (NH); 1478 (CH_2_); 1462 (N=C-S); 1438, 1580 (thiadiazole nucleus); 1070 (C-S-C); 825 (para-disubstituted benzene nucleus); 760 (C-I). ^1^H-NMR (DMSO-*d*_6_, 400 MHz), δ ppm: 4.25 (s, 2H, CH_2_); 7.28–7.32 (d, 2H, CHAr); 7.40–7.42 (d, 2H, CHAr); 8.16–8.18 (d, 2H, CHAr); 8.47–8.50 (d, 2H, CHAr); 10.27 (s, 1H, NH).

### 3.5. General Procedure for Synthesis of 3-(Benzoxazole-2′-yl-mercapto-methyl)-4-aryl-5mercapto-1,2,4-triazoles (Compounds XIV–XIX)

A total of 0.0027 moles of 1-(Benzoxazole-2′-yl-mercapto-acetyl)-4-aryl-thiosemicarbazide (II-VII) were treated with 20 mL of 2N sodium hydroxide solution at room temperature. Then, the reaction mixture was heated under reflux for 60 min. After that, the mixture was cooled, diluted with distilled water (*v/v*), and treated with a diluted solution of hydrochloric acid (1:1) to pH = 4.5. The precipitates [mercapto-1,2,4-triazole (compounds XIV–XIX) were filtered, washed on filter with 500 mL of distilled water, and finally dried. After recrystallization from boiling ethyl alcohol, the finished products were crystalline.

*3-(Benzoxazole-2′-yl-mercapto-methyl)-4-phenyl-5-mercapto-1,2,4-triazoles* (Compound XIV), Physical state: White, crystalline powder; M_p_ = 239–241 °C. FT-IR (KBr,ν, cm^−1^): 3095 (NH); 1622 (C=N); 1490 (C=S); 820 (monosubstituted benzene nucleus); 785 (S-CH_2_). ^1^H-NMR (DMSO-*d*_6_, 400 MHz), δ ppm: 4.78 (s, 2H, CH_2_); 6.99–7.01 (d, 2H, CHAr); 7.13–7.15 (d, 2H, CHAr); 7.22 (t, 1H, CHAr); 14.06 (s, 1H, NH).

*3-(Benzoxazole-2′-yl-mercapto-methyl)-4-(p-tolyl)-5-mercapto-1,2,4-triazole* (Compound XV), Physical state: White-yellow, crystalline powder; M_p_ = 198–200 °C. FT-IR (KBr,ν, cm^−1^): 3036 (NH); 1623 (C=N); 1492 (C=S); 898 (para-disubstituted benzene nucleus); 782 (S-CH_2_). ^1^H-NMR (DMSO-*d*_6_, 400 MHz), δ ppm: 2.34 (t, 3H, CH_3_); 4.51 (s, 2H, CH_2_); 7.07–7.09 (d, 2H, CHAr); 7.28–7.30 (d, 2H, CHAr); 7.51–7.53 (d, 2H, CHAr); 14.04 (s, 1H, NH).

*3-(Benzoxazole-2′-yl-mercapto-methyl)-4-(p-methoxyphenyl)-5-mercapto-1,2,4-triazole* (Compound XVI), Physical state: White-ivory, crystalline powder; M_p_ = 213–215 °C. FT-IR (KBr,ν, cm^−1^): 3100 (NH); 1628 (C=N); 1492 (C=S); 837 (para-disubstituted benzene nucleus); 783 (S-CH_2_). ^1^H-NMR (DMSO-*d*_6_, 400 MHz), δ ppm: 3.46 (t, 3H, OCH_3_); 4.29 (s, 2H, CH_2_); 7.09–7.11 (d, 2H, CHAr); 7.25–7.27 (d, 2H, CHAr); 7.53–7.55 (d, 2H, CHAr); 14.09 (s, 1H, NH).

*3-(Benzoxazole-2′-yl-mercapto-methyl)-4-(p-bromoyphenyl)-5-mercapto-1,2,4-triazole* (Compound XVII), Physical state: Brown-red, crystalline powder; M_p_ = 202–204 °C. FT-IR (KBr,ν, cm^−1^): 3328 (NH); 1617 (C=N); 1494 (C=S); 914 (para-disubstituted benzene nucleus); 780 (S-CH_2_). ^1^H-NMR (DMSO-d6, 400 MHz), δ ppm: 4.17 (s, 2H, CH_2_); 7.20–7.23 (d, 2H, CHAr); 7.47–7.50 (d, 2H, CHAr); 7.75–7.77 (d, 2H, CHAr); 14.21 (s, 1H, NH).

*3-(Benzoxazole-2′-yl-mercapto-methyl)-4-(p-chloroyphenyl)-5-mercapto-1,2,4-triazole* (Compound XVIII), Physical state: White-ivory, crystalline powder; M_p_ = 189–191 °C. FT-IR (KBr,ν, cm^−1^): 2894 (NH); 1615 (C=N); 1493 (C=S); 899 (para-disubstituted benzene nucleus); 765 (S-CH_2_); 746 (c-Cl). ^1^H-NMR (DMSO-*d*_6_, 400 MHz), δ ppm: 4.64 (s, 2H, CH_2_); 7.31–7.33 (d, 2H, CHAr); 7.42–7.44 (d, 2H, CHAr); 7.73–7.75 (d, 2H, CHAr); 14.29 (s, 1H, NH).

*3-(Benzoxazole-2′-yl-mercapto-methyl)-4-(p-iodoyphenyl)-5-mercapto-1,2,4-triazole* (Compound XIX), Physical state: Brown-red, crystalline powder; M_p_ = 222–224 °C. FT-IR (KBr,ν, cm^−1^): 3038 (NH); 1614 (C=N); 1495 (C=S); 915 (para-disubstituted benzene nucleus); 770 (S-CH_2_); 748 (C-I). ^1^H-NMR (DMSO-*d*_6_, 400 MHz), δ ppm: 4.23 (s, 2H, CH_2_); 7.35–7.37 (d, 2H, CHAr); 7.42–7.45 (d, 2H, CHAr); 7.59–7.61 (d, 2H, CHAr); 14.33 (s, 1H, NH).

### 3.6. Antibacterial Activity

The microbial strains (*Staphylococcus aureus* (ATCC-2593), *Bacillus subtilis* (ATCC-6638), *Bacillus cereus* (ATCC-10876), *Salmonella entiritidis* (P-1131) and *Escherichia coli* (ATCC-25922)) were grown on Mueller–Hinton agar, with incubation at 37 °C for 48 h. Previously, the tested substances were weighted and dissolved in dimethylsulphoxide (DMSO) to prepare the solutions with the following concentrations: 1 mg/mL, 0.5 mg/mL, and 0.25 mg/mL culture medium. The solutions thus prepared were included in 100 mL of culture medium, previously agarized, melted, and then homogenized. In parallel, the control sample was prepared consisting of glucose agar. The culture media were then distributed in the test tubes and sterilized for 20 min at 120 °C. The bacterial cultures were prepared according to the manufacture recommendations, using suspensions with a concentration of about 5.2 ⋅ 10^7^ CFU/mL.

### 3.7. Immobilization of 1,2,4-Triazole

Immobilization of the 1,2,4-triazole was performed as follows: 0.2 g powder of PG and PGB1 copolymers were introduced into 50 mL conical flasks over which 20 mL of 1,2,4-triazole solution was added (C_1,2,4-triazole_ = 3.6 × 10^−4^
**—** 15 × 10^−3^ g/mL). Then, the samples were placed in a thermostated water-bath shaker (Memmert MOO/M01, Schwabach, Germany) and shaken at 180 strokes/minute until equilibrium was reached. The 1,2,4-triazole immobilization studies on PG and PGB1 copolymers were performed at different temperatures: 25, 30, and 35 °C and for different contact times. The conical flasks were removed from the water-bath shaker, and the copolymers were centrifuged at 78 RCF for 10 min. Then, the samples were separated by filtration, and the amount of 1,2,4-triazole immobilized on PG and PGB1 copolymers was determined by UV-VIS spectrophotometry (SPEKOL 1300 spectrophotometer, Analytik Jena, Jena, Germany) at a wavelength of 226 nm, based on a calibration curve. The retained amount of 1,2,4-triazole was calculated by the difference between 1,2,4-triazole concentration in the supernatant before and after sorption process.

The amount of immobilized 1,2,4-triazole was calculated using the following equation:(12)$$q_{e}=\frac{\left(C_{0}-C_{e}\right)\cdot V}{W}\;$$
where ***q_e_*** is the amount of 1,2,4-triazole immobilized onto PG and PGB1 copolymers (mg/g), ***C***_0_ is the initial 1,2,4-triazole concentration (mg/mL), ***C_e_*** is the equilibrium 1,2,4-triazole concentration (mg/mL), ***V*** is the volume of 1,2,4-triazole solution (mL), and ***W*** is the amount of copolymers. PG and PGB1 copolymers are in the form of powder consisting of irregularly shaped particles.

### 3.8. Release of 1,2,4-triazole

In vitro 1,2,4-triazole release studies were performed by immersing the PG-T and PGB1-T systems (0.1 g) in 10 mL of simulated gastric fluids of pH = 1.2 for 2 h at 37 °C. The samples were placed in a thermostated water-bath shaker under gentle stirring (50 strokes/minute). Samples (1 µL) of supernatant solution were collected at different time intervals using a microsyringe and then analyzed spectrophotometrically at a wavelength of 226 nm, using a UV-VIS spectrophotometer (Nanodrop ND 100, Wilmington, DE, USA). The amount of 1,2,4-triazole released was determined using the calibration curve.

## 4. Conclusions

The 2-mercapto-benzoxazole molecule was used to achieve a selective support for some heterocycles with 1,3,4-thiadiazole and 1,2,4-triazole structure, respectively. The addition reaction of benzoxazolyl-2-mercapto-acetic acid hydrazide to aromatic isothiocyanantes was used to obtain a new series of acyl-thiosemicarbazides (compounds II–VII). Applying the cyclization reaction, the 4-substituted acyl-thiosemicarbazides were converted in acid medium into thiadiazole derivatives with benzoxazole residue in the molecule (compounds VIII–XIII) and in basic medium in mercapto-triazole-3,4-disubstituted (compounds XIV–XIX). All new compounds were characterized using elemental and spectral analysis (FT-IR and ^1^H-NMR). Among the active principles synthesized, 3-(benzoxazole-2′-yl-mercapto-methyl)-4-(p-methoxyphenyl)-5-mercapto-1,2,4 triazole was chosen to be immobilized onto a polymeric support due to its biological properties. The antimicrobial studies have shown that 1,2,4 triazole presents very good antimicrobial activities against several microbial strains, such as *Escherichia coli*, *Salmonella enteritidis,* and *Bacillus subtilis* compared to 1,3,4-thiadiazoles that show good antimicrobial activity against *Staphylococcus aureus*.

Two polymeric supports were used for the immobilization of 1,2,4-triazole derivative (compound XVI): grafted copolymer and grafted copolymer carrying betaine moieties based on gellan and N-vinyl imidazole, which were chosen due to the biocompatibility of the polysaccharide as well as of the polymer/grafts generated by the vinyl monomer and betaine function.

To describe the interactions that occur in sorption processes between 1,2,4-triazole and PG and PGB1 copolymers, Langmuir, Freundlich, and Dubinin–Radushkevich model isotherms were used. Kinetic sorption studies conclude that the Lagergren model best describes the experimental data and confirm that the sorption of 1,2,4-triazole on the grafting copolymers is physical in nature.

The thermodynamic study completed by the kinetic one led to the conclusion that the sorption process of 1,2,4-triazole on the betainized copolymer is more intense than in the case of the grafted copolymer only. The sorption process of 1,2,4-triazole is spontaneous and favored by the increase of temperature. The release of 1,2,4-triazole from polymeric supports proceeds through a complex mechanism controlled by both swelling and diffusion processes. Similar to the results obtained in case of immobilization of the cefotaxime sodium salt, these results confirm the potential of the grafted copolymer with betaine structure as a candidate for developing sustained/controlled drug delivery systems.

## Figures and Tables

**Figure 1 molecules-26-03330-f001:**
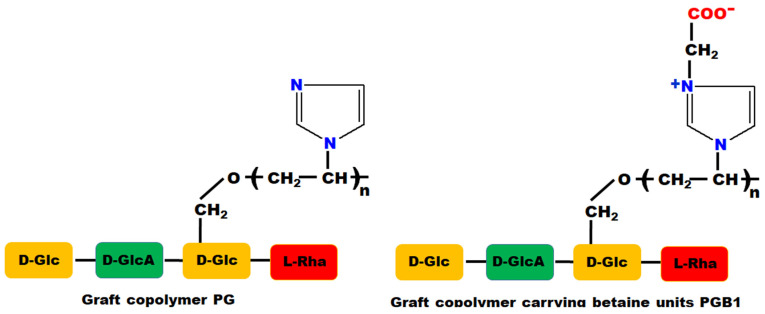
Chemical structure of PG and PGB1 graft porous copolymers.

**Figure 2 molecules-26-03330-f002:**
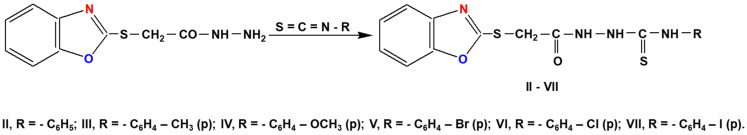
Synthesis of 1-(benzoxazole-2′-yl-mercapto-acetyl)-4-aryl-thiosemicarbazides (II-VII).

**Figure 3 molecules-26-03330-f003:**
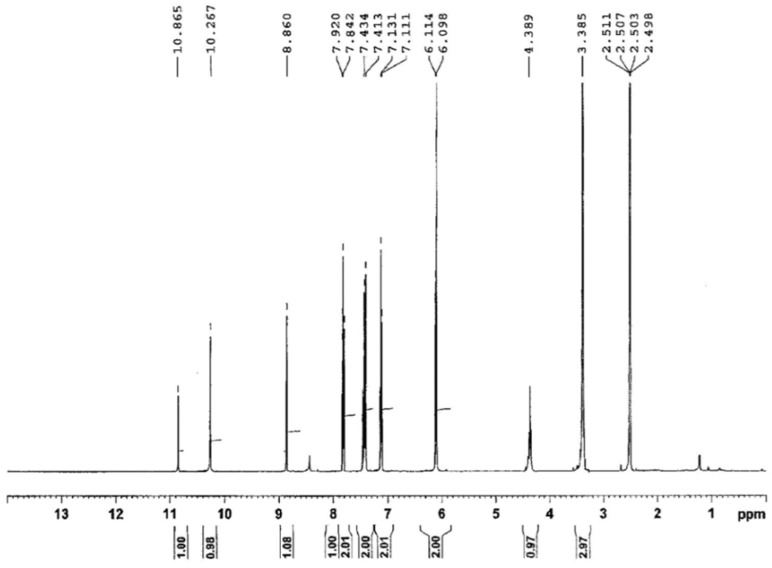
^1^H-NMR spectra of 1-(benzoxazole-2′-yl-mercapto-acetyl)-4-(p-methoxyphenyl)-thiosemicarbazide (compound IV).

**Figure 4 molecules-26-03330-f004:**
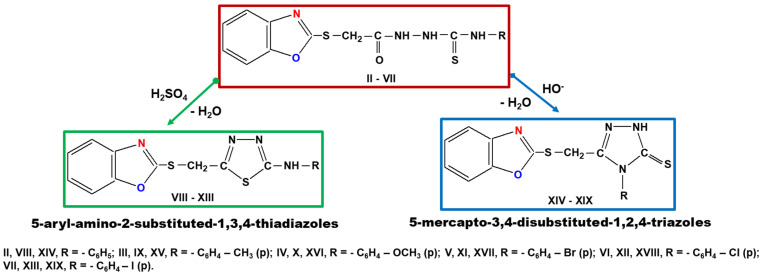
Synthesis of 1,3,4-thiadiazoles (compounds VIII–XIII) and 1,2,4-triazoles (compounds XIV–XIX).

**Figure 5 molecules-26-03330-f005:**
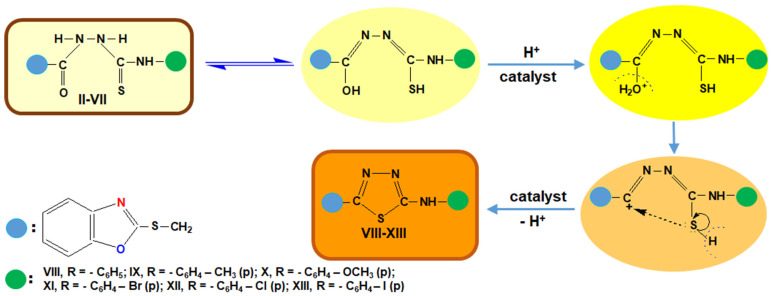
Intramolecular cyclization mechanism of reaction synthesis of 5-aryl-amino-2-substituted-1,3,4-thiadiazoles.

**Figure 6 molecules-26-03330-f006:**
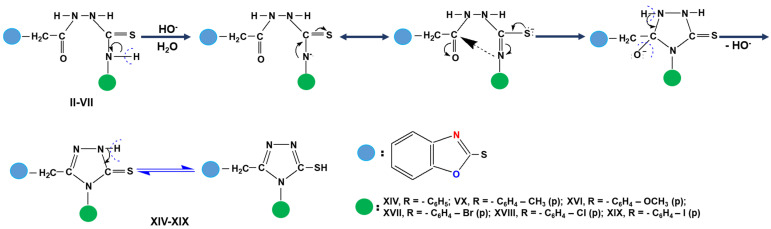
Cyclization mechanism of reaction synthesis of 5-mercapto-3,4-disubstituted-1,2,4-triazoles.

**Figure 7 molecules-26-03330-f007:**
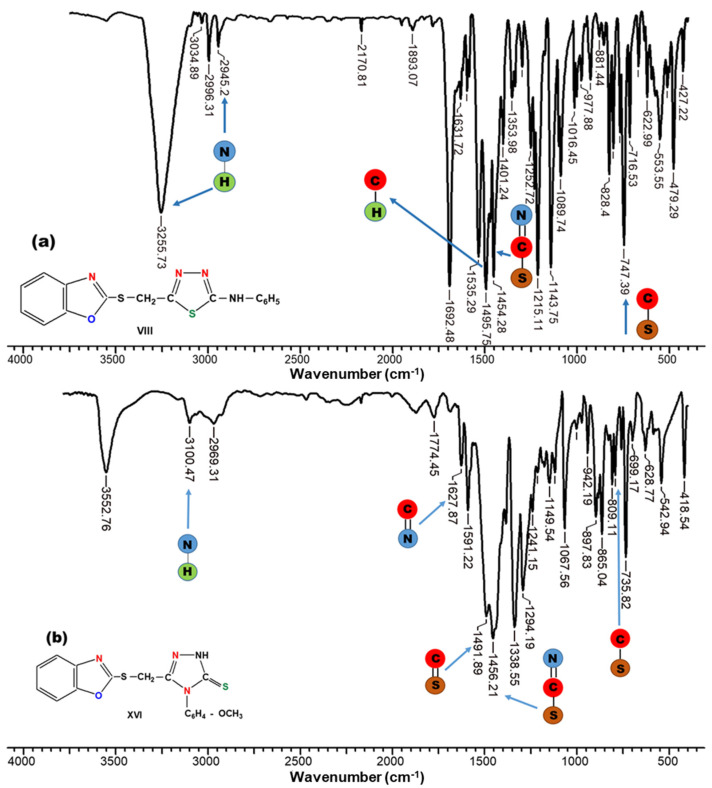
FT-IR spectrum of (**a**) 1,3,4-thiadiazoles (compound VIII) and (**b**) 1,2,4-triazole (compound XVI).

**Figure 8 molecules-26-03330-f008:**
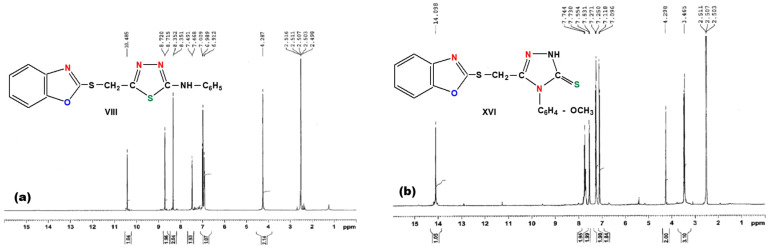
^1^H-NMR spectra of (**a**) 1,3,4-thiadiazoles (compound VIII) and (**b**) 1,2,4-triazoles (compound XVI).

**Figure 9 molecules-26-03330-f009:**
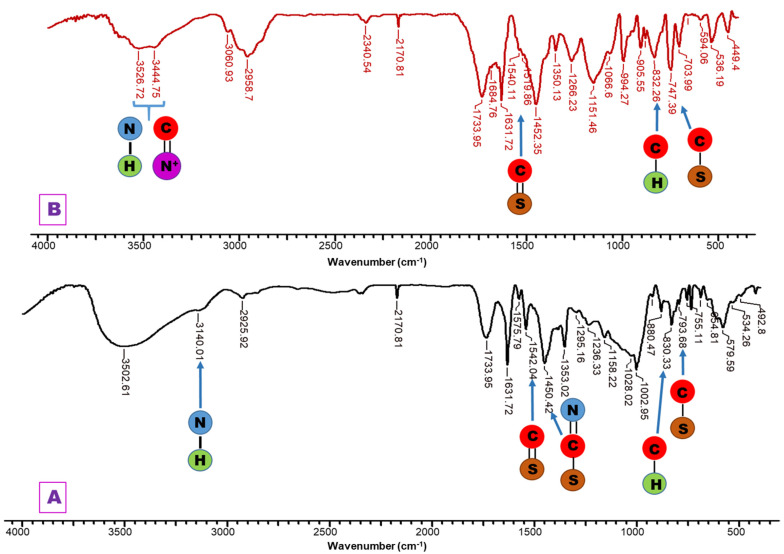
FT-IR spectra for (**A**) PG-T system; (**B**) PGB1-T system.

**Figure 10 molecules-26-03330-f010:**
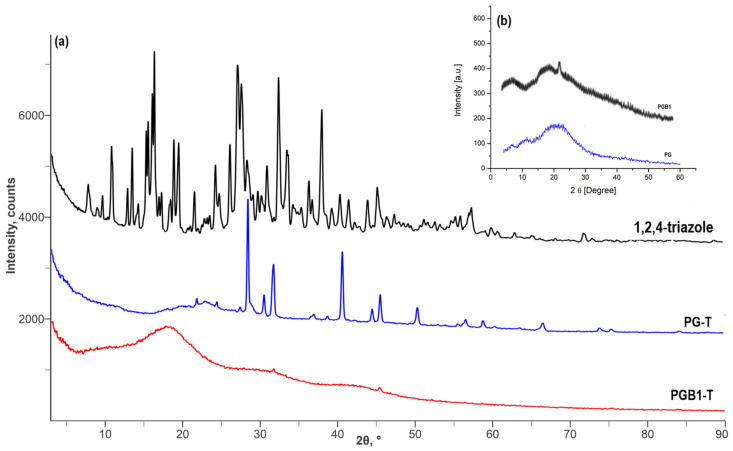
XRD patterns of 1,2,4-triazole, PG-T, and PGB1-T systems (**a**) and of PG and PGB1 copolymers (**b**).

**Figure 11 molecules-26-03330-f011:**
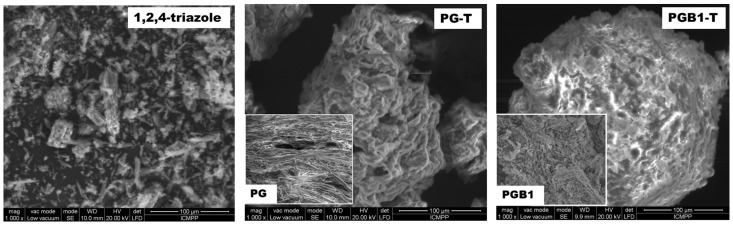
SEM images for 1,2,4-triazole, PG-T, and PGB1-T systems.

**Figure 12 molecules-26-03330-f012:**
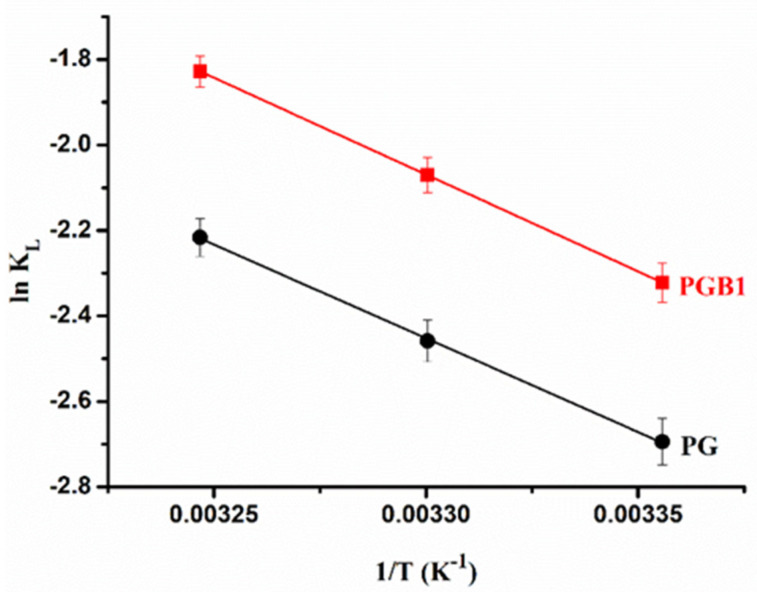
Plots of ln***K_L_*** versus 1/***T*** for sorption of 1,2,4-triazole derivative onto PG and PGB1 copolymers.

**Figure 13 molecules-26-03330-f013:**
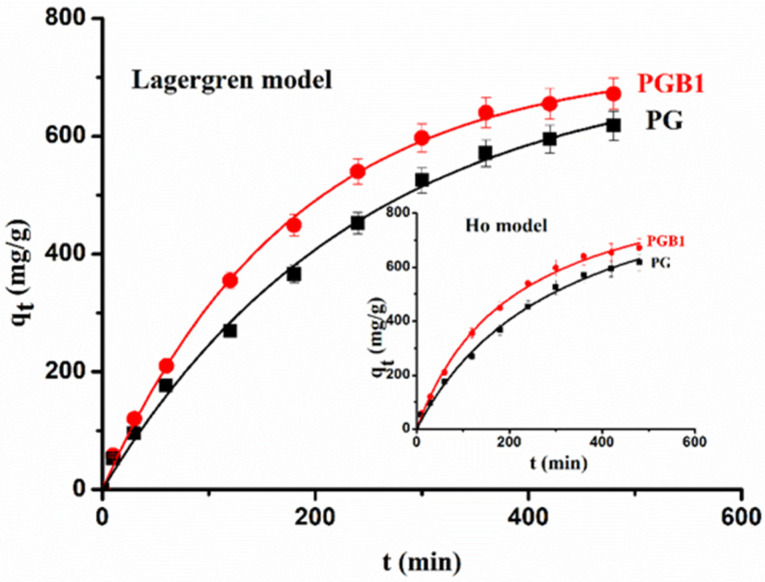
Graphical representation of Lagergren and Ho models in case of 1,2,4-triazole sorption onto PG and PGB1 copolymers at ***T*** = 308 K and C_1,2,4-triazole_ = 15 × 10^−3^ g/mL.

**Figure 14 molecules-26-03330-f014:**
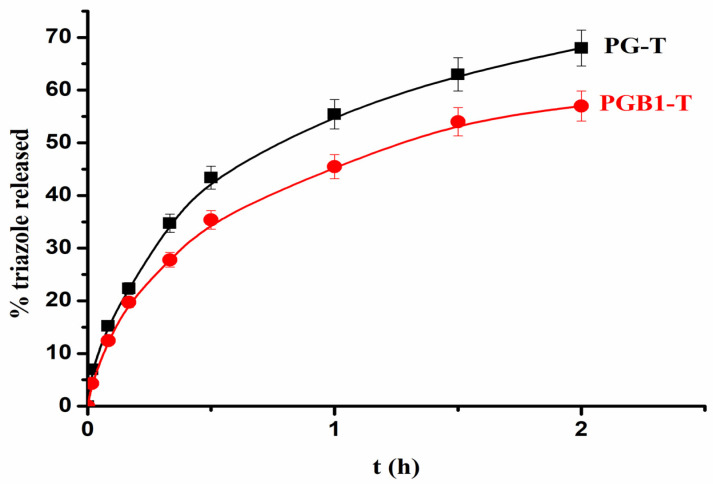
Release profiles of 1,2,4-triazole from PG-T and PGB1-T systems at pH = 1.2.

**Table 1 molecules-26-03330-t001:** Some characteristics of compounds II–XIX.

Samples	Elemental Analysis, Calc (Found) (%)								Chemical Formula	Yield (%)
	C	H	N	S	O	Br	Cl	I		
1,4-disubstituted thiosemicarbazides (compounds II–VII)										
II	53.6 (53.9)	3.9 (4.2)	15.6 (16)	17.9 (18.2)	8.9 (7.7)	-	-	-	C_16_H_14_N_4_O_2_S_2_	65
III	54.8 (55.1)	4.3 (4.5)	15.1 (15.4)	17.2 (17.6)	8.6 (7.4)	-	-	-	C_17_H_16_N_4_O_2_S_2_	75
IV	52.6 (52.9)	4.1 (4.3)	14.4 (14.8)	16.4 (16,9)	12.5 (11.1)	-	-	-	C_17_H_16_N_4_O_3_S_2_	85
V	43.9 (44.1)	2.9 (3.2)	12.8 (12.9)	14.6 (15)	7.4 (6.3)	18.3 (18.6)	-	-	C_16_H_13_N_4_O_2_S_2_Br	80
VI	48.9 (49)	3.3 (3.6)	14.3 (14.6)	16.3 (16.6)	8.2 (6.8)	-	9 (9.5)	-	C_16_H_13_N_4_O_2_S_2_Cl	78
VII	39.7 (39.9)	2.7 (2.9)	11.6 (11.9)	13.2 (13.6)	6.6 (5.2)	-	-	26.2 (26.6)	C_16_H_13_N_4_O_2_S_2_I	77
1,3,4-thiadiazoles (compounds VIII–XIII)										
VIII	56.5 (56.6)	3.5 (3.8)	16.5 (16.8)	18.8 (19.2)	4.7 (3.6)	-	-	-	C_16_H_12_N_4_OS_2_	89
IX	57.6 (58)	4 (4.1)	15.8 (16.1)	18.1 (18.5)	4.5 (3.3)	-	-	-	C_17_H_14_N_4_OS_2_	83
X	52.1 (52.4)	3.8 (4)	15.1 (15.4)	17.3 (17.6)	11.7 (10.6)	-	-	-	C_17_H_14_N_4_O_2_S_2_	81
XI	45.8 (46.2)	2.6 (2.8)	13.4 (13.6)	15.3 (15.7)	22.9 (21.7)	19 (19.4)	-	-	C_16_H_11_N_4_OS_2_Br	79
XII	51.3 (51.4)	2.9 (3.1)	15 (15.3)	17.1 (17.4)	13.7 (12.8)	-	9.5 (9.7)	-	C_16_H_11_N_4_OS_2_Cl	74
XIII	41.2 (41.5)	2.4 (2.6)	12 (12.3)	13.7 (14.1)	30.7 (29.2)	-	-	27.3 (27.5)	C_16_H_11_N_4_OS_2_I	71
1,2,4-triazoles (compounds XIV–XIX)										
XIV	56.5 (56.7)	3.5 (3.9)	16.5 (16.9)	18.8 (19.2)	4.7 (3.3)	-	-	-	C_16_H_12_N_4_OS_2_	66
XV	57.6 (57.8)	4 (4.3)	15.8 (16.2)	18.1 (18.3)	4.5 (3.4)	-	-	-	C_17_H_14_N_4_OS_2_	74
XVI	55.1 (55.3)	3.8 (4)	15.1 (15.5)	17.3 (17.7)	8.7 (7.5)	-	-	-	C_17_H_14_N_4_O_2_S_2_	88
XVII	45.8 (46)	2.6 (3)	13.4 (13.7)	15.3 (15.7)	22.9 (21.6)	19 (19.3)	-	-	C_16_H_11_N_4_OS_2_Br	79
XVIII	51.3 (51.4)	2.9 (3.3)	15 (15.2)	17.1 (17.4)	13.7 (12.7)	-	9.5 (9.9)	-	C_16_H_11_N_4_OS_2_Cl	77
XIX	41.2 (41.4)	2.4 (2.5)	12 (12.4)	13.7 (14)	30.7 (29.7)	-	-	27.3 (27.6)	C_16_H_11_N_4_OS_2_I	74

**Table 2 molecules-26-03330-t002:** Antibacterial spectra of 1,3,4-thiadiazoles and 1,2,4- triazoles.

Sample Code	Time for Thermostating (h)	*Staphylococcus aureus*			*Bacillus subtilis*			*Bacillus cereus*			*Salmonella enteritidis*			*Escherichia coli*		
		mg/mL			mg/mL			mg/mL			mg/mL			mg/mL		
		1	0.5	0.25	1	0.5	0.25	1	0.5	0.25	1	0.5	0.25	1	0.5	0.25
VIII	24	**-**	**+**	**•**	**-**	**-**	**•**	**+**	**+**	**+**	**+**	**+**	**+**	**-**	**-**	**•**
	48	**-**	**-**	**-**	**-**	**-**	**•**	**-**	**+**	**+**	**+**	**+**	**+**	**-**	**•**	**•**
IX	24	**•**	**•**	**-**	**•**	**•**	**+**	**+**	**+**	**+**	**+**	**+**	**+**	**-**	**•**	**•**
	48	**•**	**•**	**-**	**•**	**•**	**+**	**+**	**+**	**+**	**+**	**+**	**+**	**•**	**•**	**•**
X	24	**•**	**•**	**•**	**•**	**•**	**-**	**+**	**-**	**-**	**+**	**+**	**+**	**-**	**-**	**•**
	48	**•**	**•**	**•**	**•**	**+**	**-**	**+**	**+**	**+**	**+**	**+**	**-**	**•**	**•**	**•**
XI	24	**+**	**•**	**•**	**•**	**•**	**•**	**+**	**+**	**+**	**+**	**+**	**+**	**•**	**•**	**•**
	48	**+**	**•**	**+**	**•**	**•**	**+**	**+**	**+**	**+**	**+**	**+**	**+**	**•**	**•**	**•**
XII	24	**•**	**•**	**•**	**•**	**•**	**+**	**+**	**+**	**+**	**+**	**+**	**+**	**•**	**•**	**•**
	48	**•**	**+**	**•**	**•**	**•**	**+**	**+**	**+**	**+**	**+**	**+**	**+**	**•**	**•**	**•**
XIII	24	**•**	**+**	**•**	**•**	**•**	**+**	**+**	**+**	**+**	**+**	**+**	**+**	**•**	**•**	**•**
	48	**•**	**+**	**•**	**•**	**•**	**+**	**+**	**+**	**+**	**+**	**+**	**+**	**•**	**•**	**•**
XIV	24	**-**	**•**	**•**	**-**	**-**	**-**	**-**	**•**	**•**	**-**	**-**	**-**	**-**	**-**	**-**
	48	**•**	**+**	**+**	**•**	**+**	**+**	**+**	**+**	**+**	**-**	**-**	**-**	**-**	**-**	**-**
XV	24	**-**	**-**	**-**	**-**	**-**	**-**	**•**	**•**	**•**	**-**	**-**	**-**	**-**	**-**	**-**
	48	**-**	**•**	**•**	**-**	**-**	**-**	**+**	**+**	**+**	**-**	**-**	**-**	**-**	**-**	**-**
XVI	24	**-**	**-**	**-**	**-**	**-**	**-**	**•**	**•**	**•**	**-**	**-**	**-**	**-**	**-**	**-**
	48	**-**	**-**	**•**	**-**	**-**	**-**	**+**	**+**	**+**	**-**	**-**	**-**	**-**	**-**	**-**
XVII	24	**•**	**•**	**•**	**•**	**•**	**•**	**•**	**•**	**•**	**•**	**•**	**•**	**•**	**•**	**•**
	48	**+**	**+**	**+**	**+**	**+**	**+**	**+**	**+**	**+**	**+**	**+**	**+**	**+**	**+**	**+**
XVIII	24	**•**	**•**	**•**	**•**	**•**	**•**	**•**	**•**	**•**	**•**	**•**	**•**	**•**	**•**	**•**
	48	**+**	**+**	**+**	**+**	**+**	**+**	**+**	**+**	**+**	**+**	**+**	**+**	**+**	**+**	**+**
XIX	24	**•**	**•**	**•**	**•**	**•**	**•**	**•**	**•**	**•**	**•**	**•**	**•**	**-**	**•**	**•**
	48	**+**	**+**	**+**	**+**	**+**	**+**	**+**	**+**	**+**	**+**	**+**	**+**	**+**	**+**	**+**
Kanamycine	24	**-**	**-**	**-**	**-**	**-**	**-**	**-**	**-**	**-**	**-**	**-**	**+**	**-**	**-**	**-**
	48	**-**	**-**	**-**	**-**	**-**	**-**	**-**	**-**	**•**	**-**	**-**	**+**	**-**	**-**	**+**

- no growth; • moderate growth; + normal growth.

**Table 3 molecules-26-03330-t003:** Parameter values corresponding to Langmuir, Freundlich, and Dubinin–Radushkevich isotherms calculated in case of 1,2,4-triazole sorption on PG and PGB1 copolymers at different temperatures.

	PG			PGB1		
	298	303	308	298	303	308
Langmuir model						
***q_m_*** (mg/g)	478	533	636	537	616	685
***K_L_*** (L/mg)	0.068	0.086	0.109	0.098	0.126	0.161
***R_L_***	0.01–0.29	0.01–0.25	0.10–0.20	0.01–0.22	0.01–0.18	0.004–0.15
***χ*** ^2^	3.324	2.613	3.429	2.700	2.010	2.150
***R*** ^2^	0.994	0.993	0.997	0.993	0.992	0.996
Freundlich model						
***K****_F_* (L/g)	0.457	0.512	0.617	0.528	0.602	0.673
**1/** *n_f_*	0.939	0.738	0.729	0.686	0.540	0.383
***χ*** ^2^	33.462	23.958	38.332	25.213	19.219	18.782
***R*** ^2^	0.911	0.920	0.915	0.913	0.921	0.914
Dubinin–Radushkevich model						
***q_DR_*** (mg/g)	463	521	629	532	612	679
***E*** (kJ/mol)	1.099	1.561	2.171	1.357	2.608	3.714
***χ*** ^2^	0.402	0.359	0.413	0.371	0.226	0.277
***R*** ^2^	0.998	0.997	0.998	0.998	0.999	0.998

**Table 4 molecules-26-03330-t004:** Thermodynamic parameters of sorption process.

Sample Code	Δ*H* (kJ/mol)	Δ*S* (J/mol⋅K)	*R* ^2^	Δ*G* (kJ/mol)		
				298 K	303 K	308 K
PG	36.49	100.05	0.995	−29.78	−30.278	−30.78
PGB1	37.67	107.14	0.998	−31.89	−32.425	−32.96

**Table 5 molecules-26-03330-t005:** The parameters corresponding to the Lagergren and Ho models used in the case of 1,2,4-triazole sorption onto PG and PGB1 copolymers.

C_1,2,4-triazole_ (g/mL)		PG			PGB1		
		298	303	308	298	303	308
	*q_e,exp_* (mg/g)	325	394	473	412	496	587
3.6 × 10^−4^	Lagergren model						
	***q_e,calc_*** (mg/g)	346.29	415.91	491.95	428.82	511.76	600.94
	***k***_1_ (×10^3^ min^−1^)	2.19	3.01	3.31	3.83	3.92	4.14
	***χ*** ^2^	1.738	2.263	1.119	1.119	1.682	1.103
	***R*** ^2^	0.998	0.998	0.998	0.999	0.998	0.999
	Ho model						
	***q_e,calc_*** (mg/g)	449.72	482.38	564.84	579.40	616.92	696.22
	***k***_2_ [×10^6^ (g/mg⋅ min)]	1.26	1.62	2.43	2.89	3.51	3.92
	***χ*** ^2^	4.014	4.686	5.976	2.411	5.499	5.040
	***R*** ^2^	0.997	0.997	0.997	0.998	0.997	0.998
3.6 × 10^−3^	***q_e,exp_*** (mg/g)	432	491	593	507	583	659
	Lagergren model						
	***q_e,calc_*** (mg/g)	473.63	523.96	622.21	531.87	606.53	679.53
	***k***_1_ (×10^3^ min^−1^)	3.10	3.52	4.01	4.13	4.27	4.87
	***χ*** ^2^	1.456	1.446	1.569	1.312	1.852	1.786
	***R*** ^2^	0.998	0.999	0.998	0.997	0.998	0.997
	Ho model						
	***q_e,calc_*** (mg/g)	546.70	621.17	709.71	649.28	694.28	766.03
	***k***_2_ [×10^6^ (g/mg⋅ min)]	2.19	2.46	3.11	3.07	3.72	4.01
	***χ*** ^2^	2.071	2.539	2.390	2.772	2.083	2.310
	***R*** ^2^	0.997	0.996	0.997	0.995	0.997	0.996
7 × 10^−3^	***q_e,exp_*** (mg/g)	449	504	606	519	594	667
	Lagergren model						
	***q_e,calc_*** (mg/g)	463.71	527.91	622.32	533.86	621.43	681.07
	***k***_1_ (×10^3^ min^−1^)	3.42	3.96	4.43	4.60	4.90	5.10
	***χ*** ^2^	1.722	1.949	1.654	1.034	1.326	1.764
	***R*** ^2^	0.997	0.997	0.998	0.998	0.997	0.999
	Ho model						
	***q_e,calc_*** (mg/g)	567.15	613.04	713.83	621.28	734.63	796.60
	***k***_2_ [×10^6^ (g/mg min)]	3.09	3.87	4.11	3.35	4.19	5.17
	***χ*** ^2^	2.609	2.913	1.895	2.776	2.864	3.775
	***R*** ^2^	0.996	0.997	0.996	0.997	0.997	0.998
15 × 10^−3^	***q_e,exp_*** (mg/g)	457	512	618	529	601	672
	Lagergren model						
	***q_e,calc_*** (mg/g)	483.63	532.96	644.21	544.25	627.05	694.28
	***k***_1_ (×10^3^ min^−1^)	3.67	4.67	4.81	4.92	5.14	5.48
	***χ*** ^2^	1.466	1.343	1.691	1.144	1.671	1.024
	***R*** ^2^	0.997	0.999	0.998	0.998	0.999	0.999
	Ho model						
	***q_e,calc_*** (mg/g)	568.17	624.11	754.83	635.77	729.27	783.34
	***k***_2_ [×10^6^ (g/mg min)]	3.85	4.38	5.83	4.23	4.97	5.73
	***χ*** ^2^	3.050	3.208	3.903	3.887	4.321	3.678
	***R*** ^2^	0.996	0.997	0.996	0.998	0.997	0.997

**Table 6 molecules-26-03330-t006:** Kinetic parameters of 1,2,4-triazole release from PG-T and PGB1-T systems.

Sample Code	Higuchi Model		Korsmeyer–Peppas Model		
	*k_H_* (h^−1/2^)	*R* ^2^	*k_r_* (min^−n^)	*n*	*R* ^2^
PG-T	0.324	0.993	0.014	0.534	0.995
PGB1-T	0.294	0.994	0.017	0.634	0.996

## Data Availability

The data presented in this study are available on request from the corresponding author.

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
