# Peer review of "Immobilization and Release Studies of Triazole Derivatives from Grafted Copolymer Based on Gellan-Carrying Betaine Units"

_molecules, 2021, doi:10.3390/molecules26113330_

Round 1
Reviewer 1 Report
This paper describes the preparation of two new types of triazole derivatives, their antimicrobial activity, the immobilization of the derivative on two polymers, and the release of the derivative from the polymers. The results are comprehensive and contains useful information for developing drug delivery systems of immobilized drug/polymer products. However, the paper contains many equivocal points to be clarified and corrected as follows.
- There is no description on two polymers used in the present study. Probably, the detailed preparation and characterization of the polymers have been reported in a previous paper (Ref. 45). However, at least the structure of the polymers should be add in the paper, which will help readers to understand much more the content and the structure of PG copolymer (N-vinylimidazole/gellan) and PGB1 copolymer (betaine).
- Abstract: The description of the initial 6 lines may be unnecessary, because these polymers had been prepared in the previous study. If new polymers, the preparation and characterization should be added in the experimental section and in text.
- The authors have characterized the triazole derivatives by elementary analysis and IR and NMR spectra. However, mass-spectroscopic studies are necessary to characterize fully the derivatives and to obtain their molecular weights.
- Table 2: The concentration of the derivatives employed in the antibacterial studies is very high (0.25-1 mg/mL). When the derivatives are immobilized in polymers, much higher concentration may be necessary to exhibit the activity. The authors should investigate the antibacterial activity employing the immobilized products. Further, it is better to add the MIC value of the derivative or at least the derivative XVI.
- The characterization of the immobilized products is poor, only IR spectra. Other studies such as X-ray diffractometry, thermal analysis and macro- or micro-scopic observations are necessary for their full characterization. Further, the preparation method in the experimental section should be in detail described, i.e. the products were prepared by lyophilization, spray-drying or precipitation after the mixing?
- Some comments on the mechanism of the entropy-driven adsorption of the derivatives on the polymers should be added in the text, because such adsorptions usually occurs at lower temperature. Further, statistical errors should be added in the figures of Table 4.
- Table 6: the n value (0.534 and 0.634) of the Korsmeyer-Peppas model is very close to the value (0.5) of the Higuchi model. Statistical errors should be added in the figures of Table 6.
- The introduction part should be shorten and describe essential points, because it is too lengthy.
- Line 124, NMR spectra: “two ethylene protons” may be “two methylene protons”.
- Line 208: “Bacillus aureus” may be “Bacillus cereus”.
- Line 328: “the Ho model (………)” was doubly described.
Author Response
- There is no description on two polymers used in the present study. Probably, the detailed preparation and characterization of the polymers have been reported in a previous paper (Ref. 45). However, at least the structure of the polymers should be add in the paper, which will help readers to understand much more the content and the structure of PG copolymer (N-vinylimidazole/gellan) and PGB1 copolymer (betaine).
Response
Indeed, the detailed preparation and characterization methods of grafted polymers have been reported in our previous paper published in 2020 in Molecules journal (S. Racovita, N.Baranov, A.M.Macsim, C. Lionte, C. Cheptea, V. Sunel, M. Popa, S.Vasiliu, J. Desbrieres,„New grafted copolymers carrying betaine units based on gellan gum and N-vinylimidazole for immobilisation of bioactive compounds. I. Synthesis and characterization”, Molecules, 25, 5451, 2020; doi:10.3390/molecules25225451). As the reviewer suggested the structures of PG and PGB1 copolymers were added in the manuscript at the end of the Introduction Section, page 3 (Figure 1).
- Abstract: The description of the initial 6 lines may be unnecessary, because these polymers had been prepared in the previous study. If new polymers, the preparation and characterization should be added in the experimental section and in text.
Response
The Abstract has been revised and the lines regarding to the preparation methods of grafted polymers have been removed.
- The authors have characterized the triazole derivatives by elementary analysis and IR and NMR spectra. However, mass-spectroscopic studies are necessary to characterize fully the derivatives and to obtain their molecular weights.
Response
The reviewer is right and we will take into account this recommendation when characterizing other products from this class that we have synthesized in the meantime and will be the subject of a future paper. However, we specify that the products reported in this paper were purified in several stages, up to the constant of their physical properties, so we considered that it is no longer necessary to characterize by mass spectroscopy - the method to which we did not have access so far -, the physico-chemical and spectral characterization being sufficient to prove their structure and, on this basis to establish their molecular mass. The FT-IR and 1H-NMR spectra are very eloquent regarding the structure of the compounds we wanted to synthesize.
- Table 2: The concentration of the derivatives employed in the antibacterial studies is very high (0.25-1 mg/mL). When the derivatives are immobilized in polymers, much higher concentration may be necessary to exhibit the activity. The authors should investigate the antibacterial activity employing the immobilized products. Further, it is better to add the MIC value of the derivative or at least the derivative XVI.
Response
An in-depth study of the antimicrobial activity of grafted polymers, PG-T system, PGB1-T system and 1,2,4-triazole derivative as well as in vivo investigations (citotoxicity, biocompatibility, hemocompatibility) are in progress and will be the subject of a future manuscript.
- The characterization of the immobilized products is poor, only IR spectra. Other studies such as X-ray diffractometry, thermal analysis and macro- or microscopic observations are necessary for their full characterization. Further, the preparation method in the experimental section should be in detail described, i.e. the products were prepared by lyophilization, spray-drying or precipitation after the mixing?
Response
Immobilization of 1,2,4-triazole on the polymeric support was not achieved by chemical binding, this taking place through the sorption process in the pores of the grafted polymers. We used FT-IR spectroscopy only to prove that the polymer-bioactive compound system obtained by sorption contains the triazole compound.
XRD and SEM studies were performed for a better characterization of PG-T and PGB1-T systems. The PG-T and PGB1-T systems are easily separated by filtration. The supernatant was then used to determine the amount of 1,2,4-triazole absorbed onto PG and PGB1 copolymers.
- Some comments on the mechanism of the entropy-driven adsorption of the derivatives on the polymers should be added in the text, because such adsorptions usually occurs at lower temperature. Further, statistical errors should be added in the figures of Table 4.
Response
Some comments in the thermodynamic parameters as well as the R2 values were added (see pages 14 and 15).
- Table 6: the n value (0.534 and 0.634) of the Korsmeyer-Peppas model is very close to the value (0.5) of the Higuchi model. Statistical errors should be added in the figures of Table 6.
Response
The reviewer is right. Indeed, in case of PG-T system the n value of Korsmeyer-Peppas model are close to the 0.5 value of the Higuchi model. For this reason, the release mechanism is governed mainly by the diffusion process, but in case of PGB1-T system value of n parameter is higher than 0.5 and both the swelling of the matrix and the diffusion process can influence the release mechanism of bioactive compound.
- The introduction part should be shorten and describe essential points, because it is too lengthy.
Response
The Introduction Section has been shortened.
- Line 124, NMR spectra: “two ethylene protons” may be “two methylene protons”.
- Line 208: “Bacillus aureus” may be “Bacillus cereus”.
- Line 328: “the Ho model (………)” was doubly described.
Response
All corrections were made in manuscript.
Reviewer 2 Report
The authors present an interesting study in which a number of N-vinylimidazole-derived polymers are not only profiled based on a number of structural characteristics, but also functionally characterised most notably in the context of anti-bacterial and transport contexts. Detailed methodology on the synthesis of these compounds is given, and taken together, this is a comprehensive profiling which demonstrates the potential of this category of compounds for developing sustainable and controlled drug delivery systems.
In reviewing the manuscript however I had a couple of suggestions. The following should be addressed when preparing a suitable revision.
- The language for the most part is descriptive and detailed but there are sentences/sections where the language becomes almost clunky and the point being made is not clear. There are numerous instances of this, but the opening line of the introduction is a good example of this. The authors must review the entire manuscript for instances such as these in any resubmission.
- Figure 4 could be improved in terms of how it is laid out. The arrow from compound three pointing to nothing in particular should be revised.
- In table 2, the use of ‘-+’ should be considered to be replaced with something which is easier to distinguish from ‘-‘ and ‘+’.
- More details should be given on the anti-bacterial tests. For example, what concentration of bacteria was used against these compounds in determining the anti-bacterial properties of such?
Author Response
- The language for the most part is descriptive and detailed but there are sentences/sections where the language becomes almost clunky and the point being made is not clear. There are numerous instances of this, but the opening line of the introduction is a good example of this. The authors must review the entire manuscript for instances such as these in any resubmission.
Response
The English was checked and corrected by colleagues with certificate of proficiency in English.
- Figure 4 could be improved in terms of how it is laid out. The arrow from compound three pointing to nothing in particular should be revised.
Response
Figure 4 has been modified.
- In table 2, the use of ‘-+’ should be considered to be replaced with something which is easier to distinguish from ‘-‘ and ‘+’.
Response
In Table 2 the symbol “-+” was replaced with “·”.
- More details should be given on the anti-bacterial tests. For example, what concentration of bacteria was used against these compounds in determining the anti-bacterial properties of such?
Response
The concentration of bacteria cultures was added in the Method Section.
Round 2
Reviewer 1 Report
The paper has been correctly revised and is acceptable for publication.